# Cerebral Microstructure Analysis by Diffusion-Based MRI in Systemic Lupus Erythematosus: Lessons Learned and Research Directions

**DOI:** 10.3390/brainsci12010070

**Published:** 2021-12-31

**Authors:** Ettore Silvagni, Alessandra Bortoluzzi, Massimo Borrelli, Andrea Bianchi, Enrico Fainardi, Marcello Govoni

**Affiliations:** 1Rheumatology Unit, Department of Medical Sciences, University of Ferrara and Azienda Ospedaliero-Universitaria Sant’Anna, 44124 Cona, Italy; ettore.silvagni@edu.unife.it (E.S.); brtlsn1@unife.it (A.B.); 2Neuroradiology Unit, Department of Radiology, Azienda Ospedaliero-Universitaria Sant’Anna, 44124 Cona, Italy; massimoborrelli@yahoo.com; 3Neuroradiology Unit, Department of Radiology, University Hospital Careggi, 50134 Florence, Italy; dott.abianchi@gmail.com; 4Neuroradiology Unit, Department of Experimental and Clinical Biomedical Sciences, University of Florence, 50121 Florence, Italy; enrico.fainardi@unifi.it

**Keywords:** systemic lupus erythematosus, neuropsychiatric lupus, brain magnetic resonance imaging, diffusion-weighted imaging, diffusion tensor imaging, fractional anisotropy

## Abstract

Diffusion-based magnetic resonance imaging (MRI) studies, namely diffusion-weighted imaging (DWI) and diffusion-tensor imaging (DTI), have been performed in the context of systemic lupus erythematosus (SLE), either with or without neuropsychiatric (NP) involvement, to deepen cerebral microstructure alterations. These techniques permit the measurement of the variations in random movement of water molecules in tissues, enabling their microarchitecture analysis. While DWI is recommended as part of the initial MRI assessment of SLE patients suspected for NP involvement, DTI is not routinely part of the instrumental evaluation for clinical purposes, and it has been mainly used for research. DWI and DTI studies revealed less restricted movement of water molecules inside cerebral white matter (WM), expression of a global loss of WM density, occurring in the context of SLE, prevalently, but not exclusively, in case of NP involvement. More advanced studies have combined DTI with other quantitative MRI techniques, to further characterize disease pathogenesis, while brain connectomes analysis revealed structural WM network disruption. In this narrative review, the authors provide a summary of the evidence regarding cerebral microstructure analysis by DWI and DTI studies in SLE, focusing on lessons learned and future research perspectives.

## 1. Introduction

Neuropsychiatric (NP) involvement in systemic lupus erythematosus (SLE) is one of the most protean and challenging features of the disease, targeting both the central (CNS) and, to a lesser extent, the peripheral (PNS) nervous system. NP manifestations may be focal or diffuse, spanning from mild to severe clinical pictures able to heavily impact patient’s quality of life [1], and also contributing to increased mortality compared to the general population and SLE patients without NP involvement [2,3]. The real burden of neuropsychiatric lupus (NPSLE) is still unclear, lacking robust epidemiology studies, with wide heterogeneity of examined populations, different study designs, and inclusion criteria. A recent meta-analysis reported an estimated prevalence of 56.3% [4].

Regarding pathogenesis, two main pathways (not mutually exclusive) are now considered to be the most relevant. The first one, mainly related to focal NP clinical phenotypes, relies on ischaemic phenomena involving large and small blood vessels, mediated by antiphospholipid (aPL) antibodies, immune complexes, and intravascular thrombosis. The second mechanism, mainly associated with diffuse NP clinical pictures, is driven by autoimmune-mediated neuroinflammation, and involves complement activation, increased permeability of the blood–brain barrier (BBB), intrathecal migration of neuronal autoantibodies, local production of immune complexes, proinflammatory cytokines, and other inflammatory mediators [5,6,7,8,9,10].

In 1999, the American College of Rheumatology (ACR) provided a standard nomenclature and a formal set of case definitions for 19 NP syndromes (12 related to CNS and 7 to PNS) deemed as NP manifestations of SLE [11]; however, as none of them are specific to SLE, a careful process of attribution for each NP event is an essential step for diagnosis, treatment, and research [12]. Up to now, the expert clinical judgement in the context of a multidisciplinary team, along with a tight follow up of individual cases, remains the “gold standard” for diagnosis and management of NPSLE [13,14,15,16].

## 2. SLE and NPSLE: The Role of Neuroimaging

In this scenario, neuroimaging has an essential role to assist clinicians in the approach to NPSLE, mostly for differential diagnosis. Given the rapid technological progress, different tools are now available to investigate CNS involvement in SLE. Computed tomography (CT) still maintains a role in acute and emergency situations (such as haemorrhages or large infarcts) or in presence of contraindications to MRI. However, conventional brain magnetic resonance imaging (MRI) is considered the method of choice in current clinical practice to detect acute and chronic lesions [5]. In 2010, the European Alliance of Associations for Rheumatology (EULAR) has suggested a standard MRI protocol to be applied in the assessment of patients with suspected or known NPSLE, including T1/T2-weighted imaging, fluid attenuating inversion recovery (FLAIR), diffusion-weighted imaging (DWI), and gadolinium-enhanced T1-weighted sequences [17].

MRI provides detailed information on the brain and spinal cord morphology and macro-anatomy. Small punctate hyperintense T2-weighted focal lesions in subcortical and periventricular white matter (WM) and brain atrophy are the most common findings in NPSLE [18,19], but these lesions can also be observed in a high percentage of SLE patients without NP manifestations, and even in subjects without SLE. Indeed, no specific pattern of MRI brain abnormalities has been identified in NPSLE. Furthermore, around 40–50% of patients with a clinical diagnosis of NPSLE have no abnormalities on MRI [18,20,21] while, conversely, many chronic abnormalities identified by MRI in SLE patients are not associated with a specific clinically overt NP syndrome [22]. This clinical–radiologic mismatch reflects MRI underpowering to detect nervous system tissue microarchitecture or functional abnormalities.

To overcome this drawback, other advanced MRI-based imaging techniques, along with nuclear medicine modalities, can provide additional and integrative information (Table 1). A CNS biochemical profile can be explored by proton magnetic resonance spectroscopy (H1-MRS) [23,24]; nervous tissue microstructure integrity can be evaluated by diffusion-tensor imaging (DTI) [25,26] and magnetization transfer imaging (MTI) [27,28,29]; regional and whole brain perfusion can be assessed by perfusion weighted imaging (PWI) with or without contrast administration (dynamic susceptibility contrast imaging—DSC-MRI or dynamic contrast enhanced imaging—DCE-MRI, and three dimensional arterial spin labelled—3D ASL-MRI, respectively) [30]. Functional aspects such as regional brain activity and resting state networks or neuronal metabolism during tasks performance can be evaluated by different functional MRI (fMRI) modalities such as resting state fMRI (Rs-fMRI) or task-based fMRI [31,32]. Among nuclear medicine modalities, Fluorine-18 fluorodeoxyglucose-positron emission tomography (FDG-PET) explores regional brain glucose metabolism (glucose uptake and oxygen utilisation) [33,34,35] and has replaced single-photon emission computed tomography (SPECT) [36,37].

Overall, given the complexity of functional and anatomic organisation of CNS coupled with the multiplicity of the involved pathogenetic mechanisms, the heterogeneous pattern of abnormalities observed in NPSLE cannot be checked by a single neuroimaging tool and a multimodality approach by coupling conventional and advanced imaging techniques is more profitable to detect subtle or even early abnormalities, especially when conventional MRI is normal [4]. Advanced neuroimaging modalities have expanded our knowledge about the heterogeneous spectrum of tissue and functional CNS abnormalities occurring in SLE; an overview of the most recent advances in this field has been summarized by other authors [38], being beyond the scope of this review. In this article, we will focus in detail on cerebral microstructure analysis by diffusion-based MRI in SLE, with the aim to verify what we have learned and which research directions could be followed in the near future.

## 3. DWI and DTI: Physics and Clinical Application

Diffusion-weighted imaging (DWI) has become an integral part of routine neuroimaging since its introduction in the mid-1980s: it represents alterations in the random movement of water molecules in tissues, unrestricted in any direction (isotropic), revealing their microarchitecture, and it has a central role in many neurological conditions, including stroke, infections, and neoplastic processes [39]. In highly organised biological tissue, diffusion often is restricted in some directions or anisotropic: water molecules constantly encounter barriers (e.g., cell membranes, myelin fibres, and axonal projections), which prevent the water from diffusing freely. Different tissues have characteristic cellular architecture and proportions of intra and extracellular compartments: the relative proportion of the water distribution between these compartments is affected by the pathologic processes (Figure 1a–d). The apparent diffusion coefficient (ADC) is the most widely used parameter derived from the conventional DWI sequence, reflecting water motion restricted by these cell structures. It is possible to detect restriction in diffusion (decreased ADC) when there is a narrowing of the extracellular space due to an increase in cellular density caused by tumour growth (Figure 1b) or cytotoxic oedema with cellular swelling caused by acute brain ischaemia (Figure 1c). Conversely, the increase in extracellular space promoted by vasogenic oedema, demyelination (Figure 1a), or axonal loss, results in an increase in ADC.

There have been several technical improvements in DWI, leading to reduced acquisition time and artefacts, and enabling the development of diffusion tensor imaging (DTI) and other advanced techniques, such as intravoxel incoherent motion (IVIM) and kurtosis, which provide information about the complexity of tissues and their different compartments (intravascular, intracellular, and extracellular) [40]. DTI is based on the application of diffusion gradients in at least six different directions in space, enabling the evaluation of the movement of water molecules in the space and whether there is a dominant direction to diffusion restriction [41]. DTI provides several quantitative parameters, most of which are useful in the assessment of white matter conditions (Table 2). Fractional anisotropy (FA) reflects how dominant one particular water movement direction in a voxel is, providing information on the coherence of the movement direction along the WM fibres. Mean diffusivity (MD) represents a more exact value than ADC because it considers the three main directions of water movement, rendering the average motion of water molecules within the tissue. Axial diffusivity (AD) quantifies water movement along the main longitudinal direction, evaluating axonal integrity, while radial diffusivity (RD), a measure of the movement perpendicular to the primary diffusion direction, reflects myelin integrity (e.g., myelin injury leads to increased RD). Post-processed DTI scalars can be used to evaluate changes in the brain tissue caused by disease, disease progression, and treatment responses [42]. The rationale of the tractographic algorithms lies in the presupposition that in each voxel there is only one fibre population with a single fibre orientation. It follows that where more bundles of fibres coexist or where they cross, approach, converge, or diverge, the algorithm works poorly: in such areas, imaging techniques that provide higher angular resolution are needed. Intravoxel fibre crossing can be resolved using q-space diffusion imaging or a mixture model decomposition of the high angular resolution diffusion imaging (HARDI) signal [43]. The main current focus of DTI research is tracking neural fibre pathways in the central nervous system.

## 4. DWI Studies in SLE and NPSLE

In studies regarding patients with SLE, DWI abnormalities have been substantially related to the occurrence of acute ischaemic diseases, showing early ADC reduction in areas of acute/subacute ischaemia [44,45,46,47,48,49]. Therefore, the 2010 European League against Rheumatisms (EULAR) recommendations for the management of NPSLE acknowledged the role of DWI in the assessment of patients with SLE suspected of NP involvement, suggesting DWI sequences should always be performed as part of the initial MRI assessment [17]. On the contrary, increased ADC values in correspondence of T2 white-matter hyperintense lesions (WMHIs) could reflect focal vasogenic oedema in the setting of small-vessels vasculopathy [48]. Moreover, Posterior Reversible Encephalopathy Syndrome (PRES), a clinical–radiological syndrome characterised by a typical pattern of DWI alterations, has been significantly associated with SLE, despite not being specific. Even if not originally included in the 1999 American College of Rheumatology (ACR) Nomenclature [11], its prevalence has been increasingly reported, ranging rates of 0.7–1.4% among SLE patients [5,50], associated with active disease (e.g., renal and haematological involvement), hypertension, and immunosuppressive treatment. The most common MRI alteration is symmetrical, diffuse, and confluent (reversible) hyper-intensity on T2-weighted and FLAIR sequences, mostly in the subcortical WM of posterior areas of cerebral hemispheres. DWI, specifically, shows a variable degree of light hyper-/iso-/hypo-intense signal in DWI, with marked hyperintensity on ADC maps, expression of highly mobile water molecules inside vasogenic oedema.

Apart from their overt clinical utility in diagnostic settings, DWI studies were among the first ones performed in the early 2000s to approach the investigation of cerebral microstructure in SLE (Table 3), since Bosma et al. analysed normal-appearing DWI scans of 11 NPSLE patients and 10 controls [51,52]. After plotting whole-brain ADC histograms, NPSLE patients exhibited lower peak height and higher mean ADC values compared with controls, resulting from an increase in the number of voxels with higher ADC values [52]. These preliminary results have deemed the signs of decreased uniformity in the brain parenchyma, possibly due to loss of parenchymal structures, allowing interstitial water molecules to move freely in a less restricted environment. Using multi-sequence MRI (magnetisation transfer imaging, MTI, DWI, and single-volume magnetic resonance spectroscopy, MRS) in 24 NPSLE patients, the same group deepened these findings [53]. The authors demonstrated a significant correlation between increased ADC values, reflecting increased diffusivity, and decreased Magnetisation Transfer Ratio (MTR) normalised for brain volume (MTRp), a measure of homogeneity of cerebral parenchyma irrespective of differences in brain size and atrophy, suggesting patients with NPSLE have signs of loss of parenchymal structures even when conventional MRI does not detect significant atrophy.

Emmer et al. instead evaluated differences in mean ADC values at specific regions of interest (ROIs) [54]. Despite no significant differences in the grey matter (GM), WM, and hippocampus in NPSLE, non-NP SLE, and controls, the authors retrieved lower ADC values in SLE compared to controls in the amygdala, prevalent in patients with the positivity of serum N-methyl-D-aspartate receptor (NMDAR) antibodies, suggesting selective damage to the amygdala, maybe driven by autoantibodies. In 2007, Welsh et al. [49] focused on NPSLE patients with acute onset of NP manifestations (within one week from new onset of NP events deemed SLE-related). The authors, interestingly, confirmed the results of Bosma et al. in patients with past NP involvement [52]. In fact, they reported increased ADC values in NPSLE patients versus controls for the whole brain, but also at GM and WM levels, suggesting a permanent condition of reduced structural integrity, independent of acute events. The adoption of combined DWI and DTI data, exploited by Hughes et al. in 2007 [55], confirmed these findings [49,53]. In their work, the authors enrolled eight acute NPSLE females and 20 controls, reporting an increase in ADC values at the thalamus, and parietal and frontal WM, along with a coherent decrease in FA values. Consequently, DTI raised as a method of choice to study the different directions of this more isotropic movement of water molecules in the brain of SLE and NPSLE in depth.

## 5. DTI studies in SLE and NPSLE

Several studies have investigated DTI alterations in SLE patients, underlining a general pattern of reduced FA and increased MD values in SLE (both with and without NP involvement) with respect to healthy controls [55,56,57,58,59] (Figure 2), with only one report describing increased FA in the genu of corpus callosum and corticospinal tracts [60]. A recent systematic literature review [61] analysed the use of DTI in SLE and NPSLE compared to controls. In SLE, compared to controls, DTI studies showed decreased FA and increased MD in bilateral cerebral WM and grey matter. A coordinate-based meta-analysis of DTI studies demonstrated that SLE patients exhibited reduced FA compared with healthy subjects in the left striatum and bilateral inferior network, mainly affecting the corpus callosum, bilateral inferior fronto-occipital fasciculus, bilateral anterior thalamic projections, bilateral superior longitudinal fasciculus, left inferior longitudinal fasciculus, and left insula [62]. In NPSLE, a detailed decreased FA with respect to controls was described for several WM tracts, in particular corpus callosum, frontal, parietal, and temporal WM, thalamus, cingulum, fronto-occipital fasciculus, right and left uncinate fasciculus, right internal limb of internal capsule, and right corona radiata. Again, patients with NP involvement differed significantly from patients without overt NP manifestations in terms of reduced FA and increased MD values in white matter as a whole and, in addition, regional reduced FA were reported in inferior fronto-occipital and inferior longitudinal fasciculi, in the body of the corpus callosum, in the left arm of the forceps major, and in the left anterior corona radiata [56,61,63] in NPSLE patients versus non-NP SLE patients. Only two reports have not identified significant differences in MD or FA values in NPSLE compared to SLE patients without NP symptoms [57,64].

While DTI alterations were not constantly related to systemic disease activity and damage [57,58], recently, Nystedt et al. have highlighted an association between reduced FA values in the corpus callosum and disease duration in all patients with SLE, regardless of the presence of NP manifestations [57]. This observation supported the hypothesis that alteration of WM microstructure could be part of the SLE disease itself, indicating the existence of progressive WM damage in SLE even without overt NP involvement. These observations were further corroborated by three longitudinal studies in SLE. In the first one, Kozora et al. analysed 15 SLE pts over a period of 18 months [65]. A significant reduction in FA was found in the left thalamus and left cerebral white matter in the presence of stable disease activity and medications, and in the absence of changes in cognitive testing. Mackay et al., instead, employed an integrated approach, using resting-state functional FDG-PET and DTI [66]. The authors described a significant reduction in FA values in five brain areas with respect to controls. These areas included: (i) WM tracts in the parietal lobe, including a part of the superior longitudinal fasciculus; (ii) WM tracts in the vicinity of the insular, a part of the uncinate fasciculus; (iii) WM tracts in the occipital lobe/cingulum (hippocampus); (iv) WM tracts in the frontal lobe, including a part of the inferior frontal occipital fasciculus, and (v) WM tracts in the parietal lobe, including a part of the splenium of corpus callosum. FA in the para-hippocampal region correlated with high serum anti-N-methyl D-aspartate receptor antibody (anti-NMDAR ab) titres and poor performance on a spatial memory task. However, no significant longitudinal changes in microstructural integrity, measured as FA values, were reported over a mean of 14.9 months in the subgroup of 13 SLE subjects with follow-up in all 5 WM regions [66]. Recently, we have longitudinally evaluated variations of DTI metrics (FA, MD, RD, AD) in several normal-appearing WM of 17 newly diagnosed SLE patients [67]. Three normal-appearing white matter tracts exhibited statistically significant alteration between the baseline and 12 months of follow up, with a significant increase in RD and MD values in the left posterior limb of the internal capsule, an increase in RD values in the left retro-lenticular part of the internal capsule, an increase in MD in the left posterior limb of the internal capsule, and a decrease in FA in the left corticospinal tract over time. Notably, all these WM tracts are related to motor function [67]. Overall, these three studies indicated that WM microstructure alterations occurred in the absence of overt NP syndromes, starting relatively close to the diagnosis of SLE.

In conclusion, DTI alterations in patients with SLE and NPSLE may reflect the loss of WM density and structural network, with a possible worsening of these alterations in the course of the disease which seem to be irrespective of systemic disease activity and, to a lesser extent, of overt NP involvement. These findings call for better monitoring of WM tissue health, starting in the initial phases of the disease, and warrant longer follow up. Because of the lack of standardised interpretation guidelines, DTI is not currently used for clinical purposes. However, if these preliminary data will be confirmed by further studies, a reconsideration of the treatment strategy in newly diagnosed SLE can be argued to prevent further progression of WM deterioration.

## 6. Advanced DTI Applications in SLE and NPSLE: Multimodal MRI, DKI, Connectivity Matrices

Apart from DTI metrics analysis in specific WM tracts and in relation to SLE/NPSLE activity or cognitive function, DTI has been explored to assess other relevant aspects of disease pathogenesis. Since earlier studies, DTI has been combined with other MRI techniques to deepen cerebral microstructure architecture alterations, taking advantage of the different properties of the various methods simultaneously applied. As an example, the adoption of combined DWI and DTI data, performed by Hughes et al. in 2007 [55], and applied to eight acute NPSLE females and twenty controls, was forerunner to this end. As previously underlined, the authors demonstrated a decrease in FA values associated with an increase in ADC values at thalamus, parietal, and frontal WM, highlighting the coherent information derived by the two methodologies. In 2014, a group of researchers from Wroclaw Medical University [56] performed single-voxel MRS, DSC-PWI, and DTI studies in NPSLE (both active and inactive), non-NP SLE and controls with normal appearance of the brain in the standard MRI examination. Despite any significant hypoperfusion neither in the WM nor in the GM, SLE and NPSLE subjects had reduced NAA/Cr values in the posterior cingulate gyrus (PCG), with reduced FA values within a wide range of WM fibres, mostly in NPSLE, suggesting selective damage to WM fibres resulting in neuronal-axonal density reduction. Zivadinov and coworkers [68] combined volumetric measures determined on 3D T1-weighted sequences with DTI and MTI in a prospective study of diffuse NPSLE patients and controls. Diffuse NPSLE displayed reduced normalised cortical volume (NCV), and increased AD in normal-appearing (NA) WM, hypothesising a decrease in axonal density, while MTR did not significantly differ between study groups. Volumetric analyses combined with DTI were also performed to detect the association between reduced FA values and diminished WM volumes, in particular at corpus callosum (CC) [69] and frontal WM [58] levels. A technique derived from DTI is Diffusional Kurtosis Imaging (DKI), which assesses the non-Gaussian diffusion of water molecules, named diffusional kurtosis, especially in GM. To the best of our knowledge, only one study has assessed DKI in the context of SLE [70], combining it with volumetric analysis and MRS. These authors demonstrated a significant reduction in mean kurtosis at the posterior cingulate gyrus level, paired with decreased GM volume and reduction in NAA concentrations (SLE patients versus controls), suggesting a heavy impact on cognitive function of SLE patients, either with or without overt signs of NP involvement. Recently, the quantification of free extracellular water, measured exploiting a free-water (FW) imaging technique derived from DTI, confirmed an increase in the amount of extracellular free water in SLE patients with respect to healthy subjects [71]. Other researchers, instead, combined MTI with DTI, enrolling active NPSLE subjects, as well as non-NP SLE and controls [59]. Significantly higher RD, AD, and MD, as well as lower FA and MTR values, were found in NPSLE than both controls and SLE patients, with only partial overlap between the two methods. Therefore, different pathophysiologic mechanisms might be responsible for SLE pathogenesis in the context of active disease, and the inter-relationship between the processes might be captured by multimodal MRI approaches. This was confirmed in particular when the same group exploited a sophisticated technique of diffusion-weighted magnetic resonance spectroscopy (DW-MRS) [72], which combines the cell-type neuro–biochemical specificity of MRS with the microstructural sensitivity of DWI, allowing studying cellular- and compartment-specific properties of tissue microstructure by probing the diffusion of intracellular brain metabolites. Taking advantage of ultrahigh field (7T), the authors compared NPSLE, non-NP SLE subjects (*N* = 13 and 16, respectively), and controls (*N* = 19), demonstrating that the average intracellular diffusivity of different markers of glial cells metabolism, e.g., choline-containing compounds and creatine–phosphocreatine, was increased in SLE versus controls, with a linear correlation with disease activity (SLEDAI-2K). The conclusion drawn by the authors was that glial metabolism is significantly affected in SLE, and this affection might reflect the grade of systemic disease activity.

Finally, Mackay et al. [66] combined DTI data with resting-state functional FDG-PET in SLE subjects with no history of NP involvement. Resting hyper-metabolism was enhanced in the hippocampus, orbitofrontal and sensorimotor cortices, putamen/globus pallidus/thalamus, and occipital and temporal lobes, while clusters of reduced FA values were found in WM tracts adjacent to the hypermetabolic regions, such as the parietal lobe, the cingulum (hippocampus), and the splenium of the CC. The authors hypothesised an insult that might have taken place in the hypermetabolic grey matter regions, followed by disruption of the WM tracts connecting them.

Another relevant property of DTI, as suggested by these studies [66], was to evaluate the state of health of different WM tracts connecting different areas of the brain. Connectivity matrices can be derived, assessing brain connectomes in relationship to different disease characteristics [73,74,75,76]. These studies, mainly performed in the context of no NPSLE history, highlighted that connectivity matrices are altered in SLE subjects, relating to SLE-induced damage, cognitive function [73,75], and WMHIs clusters [74], and not to disease activity [73,76] or aPL and NP status [76]. The conclusions retrieved inform on a global compensatory neuroplasticity process in SLE, present even in the absence of overt NP activity. This could be the expression of the response to a subclinical diffuse disruption of WM structural integrity, a process which theoretically needs to be intercepted to limit cognitive dysfunction progression. In fact, despite structural disruption of connectivity, global preservation of functional connections remains, as demonstrated by fMRI [76], and this finding might help to prevent, at least at initial stages, a severe clinical disability.

## 7. Research Perspectives for Diffusion-Based Studies in SLE

Since DTI analysis is not without sources of biases or potential systematic errors, in particular in the determination of the b matrix value and, sequentially, on DWI experiments, other DTI-derived techniques are virtually able to overcome its technical limitations. The subjective value of the outputs derived from diffusion-weighted techniques, in particular in cerebral areas where bundles of fibres cross or merge, can be addressed by diffusion spectrum magnetic resonance imaging (DSI) tractography [77] or Q-ball [78], which are able to manage complex distributions of intravoxel fibres orientation not only in the WM but also in the cerebral and cerebellar cortex. DSI tractography measures the diffusion function directly by sampling the diffusion signal on a three-dimensional Cartesian volume. It describes diffusion in each voxel using a general function, the probability density function (PDF), which for each voxel specifies the 3D distribution of microscopic displacements of MR-visible spins that it contains. Q-ball, instead, is sensitive predominantly to the angular part of the PDF. The application of these approaches is a fundamental step toward non-invasive evaluation of connectional anatomy. Other critical points for DTI application are in areas with an excess of free water. Free water elimination model with explicit account of T2 attenuation (FWET_2_) [79] is a technique that evolves from a two-compartment model, with one of the compartments referring to free water and the other to tissue. The free water compartment is characterised by isotropic diffusion (e.g., diffusion coefficient equal to that of free water), while the tissue compartment is characterised by restricted diffusion, described by a diffusion tensor. The addition of a T2 attenuation term to each of the compartments leads to a more robust estimation of the model parameters. Such an approach permits an increase in the sensitivity when analysing voxels located at the CSF-tissue interface. At present, no study has adopted these specific techniques to evaluate cerebral microstructure in patients with SLE or NPSLE, and, along with advances in technology, even studies using ultra-high-field systems are expected in the near future. Moreover, methods of improving accuracy of DTI are able to go beyond the limitations in DTI-based tractography by gradients inhomogeneity. Since the precision and the accuracy of the diffusion tensor measurement affect the quality of the information on fibre tracking, diffusion gradient inhomogeneity determines differences between the real and the measured direction of fibres. Methods such as B-spatial distribution in DTI (BSD-DTI), a calibration method that takes into consideration non-uniform magnetic field gradients, permits improvements to the tractography fidelity by allowing to correct not only the magnitude, but also the direction of the diffusion gradient [80,81].

It must be underlined that actual DTI studies in NPSLE have some pitfalls that need to be pointed out (Table 4). First, a large heterogeneity exists in features of MRI scanners adopted, technical quality of DTI data acquisition, as well as DTI data analysis methods, which prevents full generalisability and comparison of available studies in the field. Furthermore, regarding MRI data analysis, it has to be underlined that not all DTI studies report whether WMHIs were excluded from the final quantitative analysis or not [82]. Moreover, the majority of studies are performed at single qualified centres, as a relative paucity of multicentre MRI studies is one of the major limitations in NPSLE research. Second, in patients’ selection, NP events of patients included in final analyses were not always attributed to SLE, since validated attribution algorithms were only recently developed [12,83,84] and acknowledged by international guidelines [85]. In line with recently published PWI studies [86], a validated attribution algorithm should always be included in DTI studies to properly describe patients enrolled [67]. It is probably time to rethink the actual MRI research in the context of SLE, aiming at substantially enlarging the SLE cohorts included, exploiting international and multicentre collaborations, and focusing on single manifestations (or clusters of NP manifestations, such as focal versus diffuse, or inflammatory versus thrombotic or both simultaneously occurring) with respect to considering NPSLE as a unique entity. Third, most DTI studies are cross-sectional, with a paucity of longitudinal studies, and, specifically, with no data in NPSLE patients or paired with healthy controls. Since other quantitative MRI techniques, such as H1-MRS [87,88,89,90,91] and MTI [92], provided longitudinal information coupled with NPSLE activity, this approach should also be verified in the context of DTI application. Finally, it is worth noting that rapidly developing machine-learning approaches might considerably change the amount of information that can be derived by quantitative MRI studies, included DTI, coupled with biomarkers, permitting more sophisticated comparisons among studies. It is all but clear how much it is possible to change the actual research paradigm, which considers NPSLE patients as a homogeneous group in which deriving pathogenic information on the disease itself, to a more personalised approach, able to tailor pathogenic considerations into a patient-centric perspective.

## 8. Conclusions

DWI and, particularly, DTI studies have helped in describing a global loss of WM density and structural network occurring in the context of SLE, mainly—but not exclusively, in case of NP involvement—expression of a less restricted movement of water molecules inside normal-appearing cerebral WM. Moreover, they clarified that WM microstructure alterations are a relatively early alteration along the course of the disease, probably caused by the disease itself. Consistent with this observation, it might be speculated that in early stages of SLE, when the systemic inflammatory burden of the disease is still not counteracted by treatment, neuroinflammation inside the CNS (e.g., microglia and complement deposition), coupled with blood–brain barrier altered permeability [93,94], may also already take part in contributing to the development of early brain microarchitecture and neurochemical alterations. How much these subtle abnormalities, detectable by quantitative advanced MRI techniques, will translate into future morphological MRI lesions and/or will associate with the development of different clinical NP phenotypes is a matter of future research and could be unveiled by longer longitudinal observation. Up to now, the combination of MRI techniques in multimodal studies does not fully permit to learn if neuronal damage occurs earlier than WM damage, or vice versa. However, such studies have helped in understanding the interconnection among different pathogenic moments in the natural history of the disease. Further studies, aiming at increasing sample sizes and focusing on single NP manifestations or clusters of manifestations, might define the exact role of diffusion-based MRI studies in the clinical management of NPSLE and SLE patients.

## Figures and Tables

**Figure 1 brainsci-12-00070-f001:**
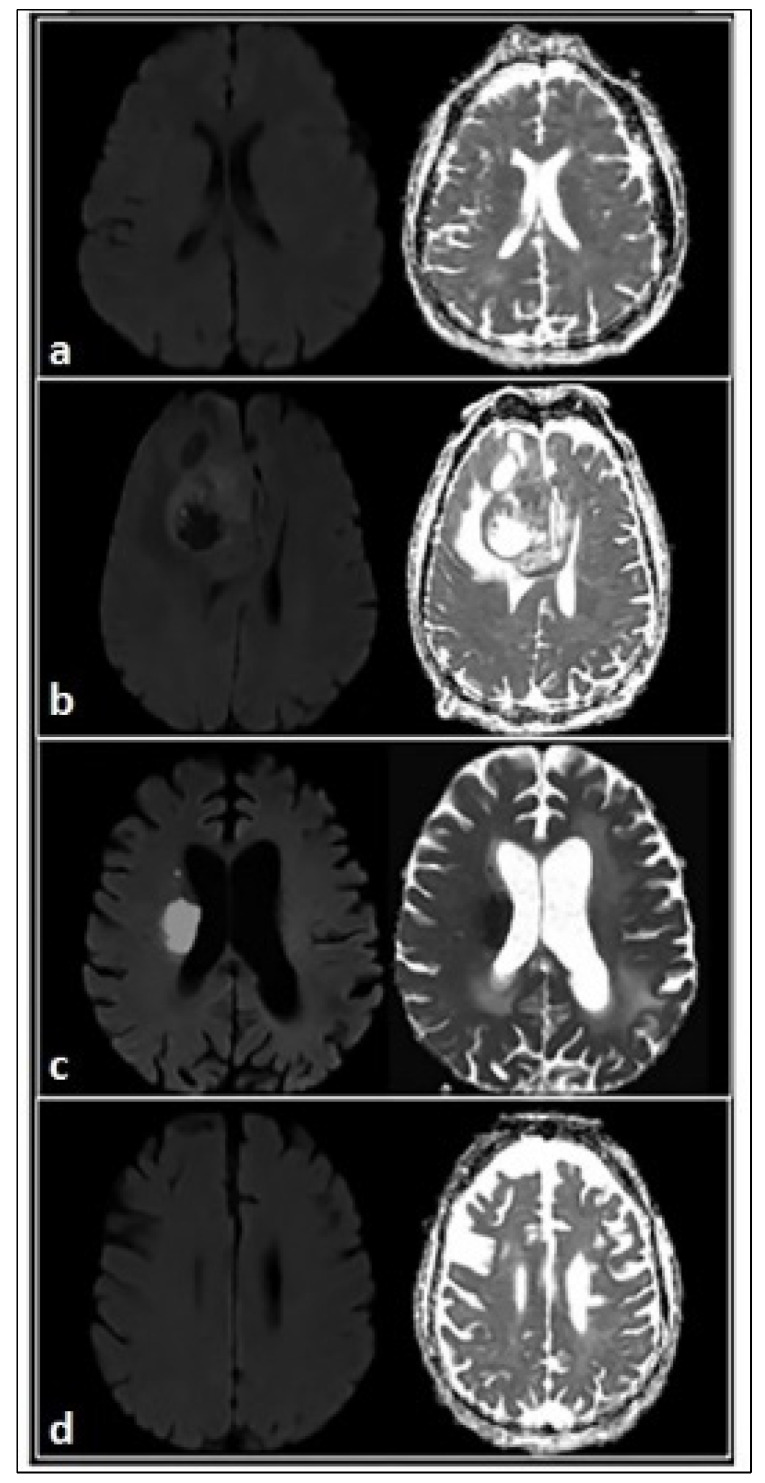
DWI technique. Restriction in diffusion (decreased ADC) occurs when there is a narrowing of the extracellular space due to an increase in cellular density caused by tumour growth (**b**) or cytotoxic edema with cellular swelling caused by acute brain ischaemia (**c**). Conversely, the increase in extracellular space promoted by vasogenic edema, demyelination (**a**), gliosis (**d**), or axonal loss, results in an increase in ADC.

**Figure 2 brainsci-12-00070-f002:**
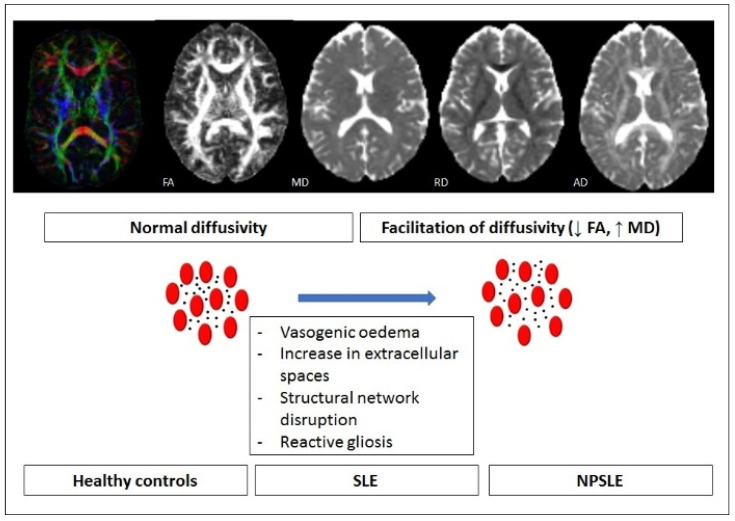
Pictorial summary of main DTI studies in the context of SLE and NPSLE. The reduction in FA and the increase in MD values in several WM tracts suggest global facilitation of cerebral diffusivity, occurring both in SLE and NPSLE. These aspects suggest different pathogenic events, such as vasogenic oedema, increase in extracellular spaces, structural network disruption, reactive gliosis.

**Table 1 brainsci-12-00070-t001:** Main MRI techniques adopted for NSPLE assessment and clinical research.

MR Imaging	Abbreviation	Explored Aspects of Nervous Tissue	Technique
Conventional brain magnetic resonance imaging	MRI	Morphology and macro-architecture	▪ T1 and T2-scans
			▪ Gadolinium-enhanced T1 sequences
			▪ Fluid Attenuated Inversion Recovery (FLAIR)
			▪ Diffusion weighted imaging (DWI)
Proton magnetic resonance spectroscopy	H1-MRS	Biochemical profile	▪ Single-voxel H1-MRS
			▪ Multi-voxel H1-MRS
Perfusion-Weighted Imaging	PWI	Brain perfusion	▪ Dynamic susceptibility contrast imaging (DSC-MRI)
			▪ Dynamic contrast enhanced imaging (DCE-MRI)
			▪ 3D arterial spin labelled (3D ASL-MRI)
Magnetisation Transfer Imaging	MTI	Brain tissue integrity of macromolecules (e.g., myelin)	
Diffusion Tensor Imaging	DTI	White matter tissue microstructure	
Functional MRI	fMRI	Neuronal connectivity and functional engagement of different brain regions	▪ Task-based functional MRI (task-fMRI)
			▪ Resting state functional MRI (rs-MRI)

Abbreviations: MRI: Magnetic Resonance Imaging; NPSLE: Neuropsychiatric Systemic Lupus Erythematosus; FLAIR: Fluid Attenuated Inversion Recovery (FLAIR); DWI: Diffusion-Weighted imaging; H1-MRS: Proton magnetic resonance spectroscopy; PWI: Perfusion-Weighted Imaging; DSC-MRI: Dynamic susceptibility contrast imaging; DCE-MRI: Dynamic contrast enhanced imaging; 3D ASL-MRI: 3D arterial spin labelled; MTI: Magnetization transfer imaging; DTI: Diffusion Tensor Imaging; fMRI: functional MRI; task-fMRI: task-based functional MRI; rs-fMRI: resting state functional MRI.

**Table 2 brainsci-12-00070-t002:** Main DTI-derived parameters and their biological significance.

DTI Parameter	Interpretation
MD	Average motion of water molecules within the tissue. It does not vary in relation to direction of the diffusion movement.
FA	Coherence of preferred water molecules movement direction along the WM fibres. It tends to negatively correlate with the degree of tissue damage. FA value approximates 0 in isotropic tissues (motion of water is equal in all directions—e.g., vasogenic oedema) and it approaches 1 in highly anisotropic environments where water is constrained to move along a primary direction (e.g., cytotoxic oedema).
AD	Rate of diffusion parallel to primary diffusion direction. It evaluates axonal integrity.
RD	Rate of diffusion perpendicular to the primary diffusion direction. It tends to increase in presence of demyelination.

Abbreviations: DTI: Diffusion Tensor Imaging; MD: mean diffusivity; FA: fractional anisotropy, AD: axial diffusivity; RD: radial diffusivity.

**Table 3 brainsci-12-00070-t003:** Studies investigating quantitative ADC alterations in SLE patients.

Author, Year	Patients	Brain MRI Techniques	Main Findings
Bosma et al., 2003 [52]	11 NPSLE (past NP involvement), 10 HCs.	DWI	NPSLE vs. HCs: lower peak height and higher mean ADC values (whole brain).
Bosma et al., 2004 [53]	24 NPSLE (past NP involvement).	DWI, MTI, MRS	Correlation between reduced MTRp and increased mean ADC values (whole brain).
Emmer et al., 2006 [54]	37 NPSLE (past NP involvement), 21 SLE, 12 HCs.	DWI	Lower mean ADC values at amygdala level in SLE and NPSLE compared to HCs, in particular for NMDAR-Ab positive patients.
Welsh et al., 2007 [49]	17 acute NPSLE (within one week from symptoms onset), 21 HCs.	DWI	Higher ADC values at whole brain, GM and WM levels for acute NPSLE compared with HCs.
Hughes et al., 2007 [55]	8 acute NPSLE (within one week from symptoms onset), 20 HCs.	DWI, DTI	NPSLE vs. HCs: increased ADC values and decreased FA at thalamus, parietal and frontal WM levels.

Abbreviations: ADC: Apparent Diffusion Coefficient, SLE: Systemic Lupus Erythematosus; MRI: Magnetic Resonance Imaging; NPSLE: Neuropsychiatric Lupus Erythematosus; HCs: healthy controls; DWI: Diffusion-Weighted Imaging; MTI: Magnetisation Transfer Imaging; MRS: Magnetic Resonance Spectroscopy; MTRp: peak height of Magnetisation Transfer Ratio normalised for brain parenchyma volume; NMDAR-Ab: anti-N-Methyl-D-Aspartate receptor antibodies; GM: grey matter; WM: white matter; DTI: Diffusion Transfer Imaging; FA: Fractional anisotropy.

**Table 4 brainsci-12-00070-t004:** Main research pitfalls regarding DTI studies in SLE and NPSLE.

Type of Limitation	Explanation
Data acquisition	Heterogeneity in MRI scanners adopted.
	Heterogeneity in DTI data acquisition.
Data analysis	Heterogeneity in DTI data analysis methods.
	Exclusion of lesions masks not always reported.
Generalisability of results	Small sample sizes.
	Lack of multicentre studies.
Patients’ selection	Heterogeneity in types on NP events or syndromes included.
	NP events not always attributed to SLE.
Study design	Paucity of longitudinal studies.
Clinical utility	Actual lack of translation into a patient-centred perspective of DTI studies.

Abbreviations: DTI: diffusion-tensor imaging; SLE: Systemic Lupus Erythematosus; NPSLE: Neuropsychiatric Lupus Erythematosus; MRI: Magnetic Resonance Imaging.

## Data Availability

Not applicable.

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
