# Peer review of "Cerebral Microstructure Analysis by Diffusion-Based MRI in Systemic Lupus Erythematosus: Lessons Learned and Research Directions"

_brainsci, 2021, doi:10.3390/brainsci12010070_

Round 1
Reviewer 1 Report
in file

Author Response
Please, see the file enclosed.

Reviewer 2 Report
Thanks to the authors for their effort. This is a nicely written review paper. One suggestion for this paper is to add an outlook or future direction section in the end. This should include something like emerging approaches beyond DTI, e.g., FWET2, DKI, DSI, Q-ball as well as using connectome gradients and ultra-high-field systems.
Author Response
Please, see the file enclosed.

Round 2
Reviewer 1 Report
Dear Authors,
The manuscript is better, carefully written, and better thought out. The fact that certain techniques (DWI) are used for a specific purpose means that we should consciously use them knowing the strengths and weaknesses. This knowledge is critical to the quality of the measurement and possible conclusions. In my opinion, the narrative of the work lacks a short description of possible systematic errors. The aforementioned BSD-DTI issue describes the scale of the problem well and shows the possible consequences and ways of solving them. In my opinion, a short mention will give the work a deeper character and a wider group of readers.
A lot of success in the New Year
Author Response
POINT BY POINT REBUTTAL LETTER
To Prof. Dr. Stephen D. Meriney,
Editor-in-Chief
Brain Sciences
Dear Sir,
Thank you for the consideration of our work and for the opportunity to revise our paper. We are grateful to the reviewers for their comments and suggestions. Please, find enclosed a detailed response to all comments raised by reviewers. All the authors have approved these revisions. The clean and marked copies of this new revised version of the manuscript are enclosed.
Reviewer 1
Dear Authors, the manuscript is better, carefully written, and better thought out. The fact that certain techniques (DWI) are used for a specific purpose means that we should consciously use them knowing the strengths and weaknesses. This knowledge is critical to the quality of the measurement and possible conclusions. In my opinion, the narrative of the work lacks a short description of possible systematic errors. The aforementioned BSD-DTI issue describes the scale of the problem well and shows the possible consequences and ways of solving them. In my opinion, a short mention will give the work a deeper character and a wider group of readers. A lot of success in the New Year.
We are grateful to this reviewer for the suggestions and we apologize if we did not focus exactly on this issue in the previous rebuttal letter. We tried to improve the quality of our work by providing further information in the chapter “Research perspectives for diffusion-based studies in SLE” (page 8) – see answer to Reviewer 2. Specifically, we addressed the issue regarding BSD-DTI by including some references at the end of the paragraph related to advanced DTI techniques and methods to overcome the known limitations of DWI and DTI.
Reviewer 2
I appreciate the authors for addressing my comments. It would be better to include further introductory information relating to the emerging techniques.
We are grateful to this reviewer for the suggestions. We tried to improve the quality of our work by providing further information in the chapter “Research perspectives for diffusion-based studies in SLE” (page 7-8). In particular, we have modified the pipeline, adding a paragraph focusing on these more advanced DTI techniques before enlisting the main limitations of DTI studies in SLE and NPSLE.
Reviewer 2 Report
I appreciate the authors for addressing my comments. It would be better to include further introductory information relating to the emerging techniques.
Author Response

(The authors gave the same response as above.)
